# Grain Boundary Precipitation Control of GCP Phase Using TCP or A2 Phase in Ni-Based Alloys

Shuntaro Ida [1],*,[†], Ryosuke Yamagata [2], Hirotoyo Nakashima [2], Satoru Kobayashi [2] and Masao Takeyama [2]

1    Department of Metallurgy and Ceramics Science, Graduate School of Engineering,
     Tokyo Institute of Technology, 2-12-1-S8-8 Ookayama, Meguro-ku, Tokyo 152-8552, Japan
2    Department of Materials Science and Engineering, School of Materials and Chemical Technology,
     Tokyo Institute of Technology, 2-12-1 Ookayama Meguro-ku, Tokyo 152-8552, Japan
*    Correspondence: shuntaro.ida.e1@tohoku.ac.jp
†    Current Address: Department of Materials Science, Graduate School of Engineering, Tohoku University,
     6-6-02 Aramaki Aza Aoba, Aoba-ku, Sendai 980-8579, Japan.

**Abstract:** To cover the grain boundary (GB) of the Ni phase with precipitates, the GB precipitation behavior of both topologically close-packed (TCP) or A2 and geometrically close-packed (GCP) phases was investigated in two Ni–Nb–(Co, Cr) ternary systems. The Ni/TCP or A2/GCP three-phase region existed in both systems. In the Ni-Nb-Co ternary system, Nb was approximately equally partitioned into both $Co_7Nb_2$ ($mC18$ structure, TCP) and $(Ni, Co)_3Nb$ ($D0_{19}$ structure, GCP) phases. In the Ni–Nb-Cr ternary system, Nb and Cr were mainly partitioned into the $Ni_3Nb$ ($D0_a$ structure, GCP) and Cr (A2 structure) phases, respectively. In the Ni–Nb–Co ternary system, the $Co_7Nb_2$ phase grew along the GB, whereas the $(Ni, Co)_3Nb$ phase grew toward the grain interior (GI). However, the growth of the $Ni_3Nb$ phase toward the GI was suppressed in the Ni–Nb–Cr ternary system. The suppression of growth of the GCP phase and covering the GB using both the TCP or A2 and GCP phases might be possible in a system where the precipitation of the GCP phase nucleating on the GB prior to the TCP or A2 phase increases supersaturation for precipitation of the TCP or A2 phase.

**Keywords:** geometrically close-packed phase; topologically close-packed phase; grain boundary precipitation





## 1. Introduction

Improving the energy efficiency of fossil-fuel power plants is critical for alleviating energy and environmental problems. Various R&D projects related to novel high-temperature materials applicable to 700 °C-class advanced ultra-supercritical (A-USC) power plants are underway [1–4].

The material for boiler tubing in next-generation power plants is required to have long-term strength (973 K, 100 MPa, $10^5$ h creep rupture strength). Promising candidate materials for boiler tubing include HR6W, alloy 617, alloy 625 and alloy 740/740H [5–11]. With the expected increase in steam temperature, the turbine inlet temperature for next-generation power plants is also scheduled to increase. The durable temperature of the material for the rotor disk must also be improved. A well-known superalloy, Inconel 718, which is widely used in aircraft, power plants and the chemical industry, has also been studied to expand the operation range [12,13]. These wrought Ni-based alloys are basically strengthened with a geometrically close-packed (GCP) phase, such as the $\gamma'$ phase ($Ni_3Al$) or $\gamma''$ phase ($Ni_3Nb$), within the grain interior (GI). The GCP phase is a derivative of the Ni phase because the atom arrangement on the close-packed plane of the phase is the same as that of the Ni phase if the difference in atoms is disregarded. Therefore, the GCP phase precipitates coherently within the GI and an increase in the volume fraction of the phase strengthens the Ni-based alloys. Microstructure design can effectively improve the relatively short-term (~1000 h) strength [14]. However, for materials used in power-plant

applications that require long-term strength, new microstructure design concepts for the grain boundary (GB) are needed.

Takeyama et al. used Ni–Cr–W model alloys in which the $\alpha_2$-W (A2 structure) phase covers the GB to clarify that decorating the GB with a thermodynamically stable phase improves creep resistance [15,16]. They found that the minimum creep rate decreases with an increase in the area fraction of the GB covered with precipitates, and they named the strengthening mechanism "grain boundary precipitation strengthening" (GBPS). This GBPS mechanism has also been confirmed in novel heat-resistant austenitic steels in which thermodynamically stable topologically close-packed (TCP) and GCP phases with relatively high coherency with the Ni phase are precipitated at the GB and within the GI, respectively. The austenitic steels meet the requirements for boiler tubing of A-USC power plants [17–19]. Therefore, a microstructure design in which the GB is covered with thermodynamically stable phases at high temperatures is expected to be important for improving long-term strength, even in the next-generation wrought Ni-based alloys. However, the GB design by GCP phases of the currently developed wrought Ni-based alloys is not sufficient. For example, alloy 740, which has better creep strength than HR6W and alloy 617, forms the MC and $M_{23}C_6$ carbides and the $\gamma'$ phase at the GB. However, the $\gamma'$ phase transforms into the stable η phase, and the coarsening G phase is also formed during aging and creep [8–10]. Alloy 740H was developed to improve the thermomechanical stability and to suppress the formation of the η and G phases. [10,20]. However, the coarsening of carbide and $\gamma'$ phases at the GB remains a problem because of the formation of the uncovered GB, where creep voids preferentially form [8]. Inconel 718 forms one of the GCP phases of the $Ni_3Nb$ phase ($D0_a$ structure, GCP phase), which is the thermodynamically stable phase of the $\gamma''$ phase [21]. Its role is, however, limited to a pinning effect for controlling the grain size. Some authors have reported the precipitation of the $Ni_3Nb$ phase at the GB. Gallo et al. reported that random high-angle boundaries are more populated with the phase than low coincidence site lattice (CSL) special boundaries and that nonspecial triple junctions are also preferential nucleation sites compared with 2- or 3-CSL triple junctions [22]. Other authors who have reported on precipitation of the phase found that the $Ni_3Nb$ phase can also preferentially nucleate at the GB but then grows toward the GI in the $Ni/Ni_3Nb$ two-phase region [23]. The area fraction of a GB covered by the precipitate ($\rho$) differs substantially by GB (i.e., between 30% and 100%), whereas the average area fraction ($\bar{\rho}$) increases to ~75% after long-term aging. The observed large difference in $\rho$ is found to be enhanced in the precipitation growth stage. A crystallographic orientation analysis indicates that the difference might be caused by the geometry of the habit planes, $\{111\}_{Ni}$, with respect to the GB plane; that is, in a case where one of the habit planes is nearly parallel to the GB plane, the $Ni_3Nb$ phase grows along the GB, resulting in high $\rho$, while the phase grows towards the GI when any of the habit planes are inclined toward the GB plane, resulting in low $\rho$. This phenomenon is caused by the GCP phase's structure being a derivative of the Ni phase's structure and makes covering every GB with only the GCP phase difficult.

Our novel microstructure design concept for wrought Ni-based alloys is to use TCP or A2 phases to cover GBs and use the GCP phase to strengthen the alloy by coherent precipitation within the GI. The TCP and A2 phases can preferentially nucleate and grow along the GB [15–19,24,25], which may be due to their relatively different structures compared to the GCP phase. The TCP phase consists of relatively large and small atoms. The small atoms are arranged to form a close-packed tetrahedron. The large atoms occupy a relatively large space between the tetrahedra. The atom arrangement of planes of the TCP phase, such as the Kagome-net and $3^6$-net, is different from that of the close-packed plane of the Ni phase [26–28]. For example, the Kagome-net has an atom arrangement in which atoms are regularly taken from the close-packed plane of the Ni phase. Therefore, unlike the GCP phase, the TCP and A2 phases are not derivatives of the Ni phase, and it is difficult for them to grow toward the GI according to their crystal orientation relationship with the Ni phase. As a result, they grow along the GB with high energy and can easily cover the GB.

However, independent precipitation of the TCP or A2 phase on the GB and precipitation of the GCP phase within the GI could be difficult in terms of precipitation kinetics on the GB. According to nucleation theory, all precipitates tend to nucleate preferentially at the GB because the GB energy reduces the activation energy for nucleation [29]. The precipitation kinetics on the GB for the GCP phase would be much faster than those for the TCP and A2 phases because of the crystal structure of the GCP phase is a derivative of that of the Ni phase. When both GCP and TCP or A2 phases are used as strengthening phases in Ni-based alloys, both TCP or A2 and GCP phases may precipitate at the GB.

Therefore, the precipitation behavior of the TCP or A2 phase and the GCP phase at the GB was investigated using model alloys in two types of Ni–Nb–*M* ternary systems. The Ni/TCP or A2/GCP three-phase region exists in both systems, Type I, in which Nb is approximately equally partitioned into TCP and GCP phases, and Type II, in which Nb and M elements are mainly partitioned into the GCP and A2 or TCP phases, respectively. The Ni-Nb-(Co, Fe, V) and Ni-Nb-(Cr, Mo, W) ternary systems are examples of Type I and Type II, respectively [15,30–33].

The objective of this study was to investigate the GB precipitation behavior of both the TCP or A2 phase and the GCP phase to clarify the GB design principle using these phases.

## 2. Materials and Methods

Ni–Nb–Co and Ni–Nb–Cr were used as the ternary systems for Type I and II, respectively. The Ni–Nb–Co ternary system had a Ni/$Co_7Nb_2$ (*m*C18 structure, TCP phase)/(Ni, Co)$_3$Nb (D0$_{19}$ structure, GCP phase) three-phase region [30], and the Ni–Nb–Cr ternary system had a Ni/Cr (A2 structure)/Ni$_3$Nb three-phase region [32].

The alloys were prepared as ~30 g button ingots by arc melting using a nonconsumable tungsten electrode. Each ingot was melted five times and was turned over each time to avoid segregation. The specimens were cut into pieces by electro discharged machining.

The alloys used in the Ni–Nb–Co ternary system (Type I) were Ni-5Nb-80Co and Ni-6Nb-57Co. The composition of the alloys was decided on the basis of the isothermal section at 1373 K and 1473 K [30]. The isothermal section suggests that all the alloys were in the Ni single-phase region at 1473 K and that Ni-5Nb-80Co and Ni-6Nb-57Co would be in the Ni/$Co_7Nb_2$ two-phase region and Ni/$Co_7Nb_2$/(Ni, Co)$_3$Nb in the two-phase region, respectively, at 1273 K. The specimens were cold-worked to 30–40%. The solution was treated at 1473 K for 4–24 h to achieve a grain size of ~150 μm, and subsequently aged at 1273 K for 1–1000 h, followed by water quenching.

The alloys used in the Ni–Nb–Cr ternary system (Type II) were Ni-15Nb-(30,50)Cr and Ni-3.3Nb-39.4Cr. The former alloys were used for investigating changes in the Ni/Cr/Ni$_3$Nb three-phase region with changes in temperature and were equilibrated at 1373 K/240 h and 1273 K/1344 h, followed by water quenching. The equilibration heat treatment time was determined by the diffusion coefficient of Nb and Cr, which can diffuse by more than 100 μm in pure Ni [34,35]. The latter alloy was used for investigating the precipitation behavior of the Cr and Ni$_3$Nb phases at the GB and was cold-worked to 30–40%. The solution was treated at 1423 K for 12 h to achieve a controlled grain size of ~150 μm, and subsequently aged at 1273 K for 1–1000 h, followed by water quenching.

All heat treatments were performed in a silica tube backfilled with Ar gas after being evacuated at $2.7 \times 10^{-3}$ Pa. The microstructures were examined using field-emission scanning electron microscopy (FE-SEM) and transmission electron microscopy (TEM). TEM discs with a thickness of 0.1 mm and diameter of 3 mm were machined and then mechanically polished, followed by twin-jet electropolishing in an ethanolic solution of 10 vol% perchloric acid at approximately 248 K. For the powder X-ray diffraction (XRD) analyses, samples were crushed and particles with a diameter smaller than 45 μm were used. The powder was annealed for 10 min at the heat treatment temperature that was reached prior to powdering to remove the strain introduced during crushing. Measurements were performed under the following conditions: target, Cu (K$\alpha_1$, $\lambda = 1.5406$ Å); voltage, 40 kV; current, 40 mA; divergence slit, 2/3°; scattering slit, 8.0 mm; scanning speed,

$3.00°$/min; and step width, $0.02°$. The phase composition was analyzed by an electron probe microanalyzer (EPMA) equipped with a wavelength-dispersive X-ray spectroscope under operating conditions of 20 kV and $2.0 \times 10^{-8}$ A and was determined for more than five data sets calibrated using the ZAF correction method with pure elements as standards.

For the quantitative evaluation of GB precipitation, two area fractions were measured [23,25]. The first was the area fraction of precipitates at each individual GB (hereafter designated by $\rho$). This fraction was calculated using the following equation:

$$\rho_i(\%) = \sum_k l_{ik}/L_i \tag{1}$$

where $L_i$ is the length of GB $i$ and $l_{ik}$ is the length of precipitate $k$ at GB $i$.

The second type was the average area fraction ($\bar{\rho}$), which was calculated by the following equation:

$$\bar{\rho} = \sum_i \sum_k l_{ik} / \sum_i L_i \tag{2}$$

The length of the GB and precipitate at the GB were measured with an opisometer and a caliper, respectively.

## 3. Results

### 3.1. Precipitation of $Co_7Nb_2$ Phase at the GB

The precipitation behavior of the TCP phase at the GB is shown using the $Co_7Nb_2$ phase in Ni-5Nb-80Co. Figure 1 shows the change in microstructure with aging of the specimen after the solution treatment at 1473 K for 24 h. The $Co_7Nb_2$ phase with a granular morphology preferentially nucleated at the GB after the aging treatment at 1373 K for 10 h, and growth toward the GI was not observed (Figure 1a). After the aging treatment at 1273 K for 10 h (Figure 1b), the same GB precipitation tendency was observed. In addition, the $Co_7Nb_2$ phase with a Widmanstätten morphology was observed within the GI. After the aging treatment at 1273 K for 100 h (Figure 1c), although the volume fraction of the $Co_7Nb_2$ phase increased, growth toward the GI was still not observed. After the aging process at 1173 K for 100 h (Figure 1d), part of the GB $Co_7Nb_2$ phase precipitated in a discontinuous manner and the TCP phase grew toward the GI. Therefore, excessive supersaturation causes the discontinuous precipitation and growth of the TCP phase toward the GI.

Figure 2 shows TEM images of a grain of Ni-5Nb-80Co after the specimen was aged at 1273 K for 100 h. In the dark-field image taken with $g = 001_{Co7Nb2}$, the $Co_7Nb_2$ phase with a plate-like morphology was observed within the GI (Figure 2a). The interface was straight. In the diffraction pattern corresponding to the area indicated by a dashed line in Figure 2a, the intensity of the $Co_7Nb_2$ phase was weaker than that of the Ni phase (Figure 2b). The patterns of two variants of the $Co_7Nb_2$ phase were discernible. Both $(001)_{Co7Nb2}$ were parallel to $\{111\}_{Ni}$. Therefore, a crystal orientation relationship existed between the Ni and $Co_7Nb_2$ phases ($\{111\}_{Ni} //(001)_{Co7Nb2}$, $<1\bar{1}0>_{Ni} //[010]_{Co7Nb2}$) and the $Co_7Nb_2$ phase within the GI precipitates on the habit plane, $\{111\}_{Ni}$. The GB $Co_7Nb_2$ phase was thicker than the GI $Co_7Nb_2$ phase, and the interface was not straight (Figure 2c). In the diffraction pattern corresponding to the area indicated by a dashed line in Figure 2c, the patterns of the Ni phase and one variant of the $Co_7Nb_2$ phase were detected (Figure 2d). The pattern of the GB $Co_7Nb_2$ phase was the same as the pattern of the variant of the GI $Co_7Nb_2$ phase. Even if the orientation was the same, the GB $Co_7Nb_2$ phase and GI $Co_7Nb_2$ phase precipitated along the GB and on $\{111\}_{Ni}$, respectively.

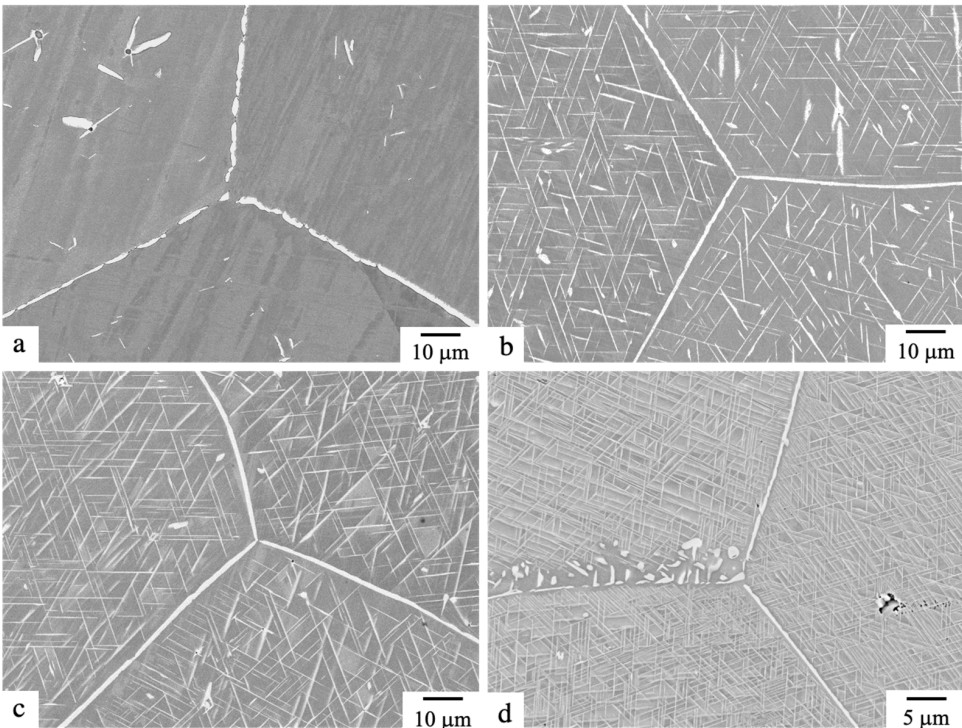

**Figure 1.** Backscattered electron images of Ni-5Nb-80Co after solution treatment followed by aging at (**a**) 1373 K for 10 h, (**b**) 1273 K for 10 h, (**c**) 1273 K for 100 h and (**d**) 1173 K for 100 h.

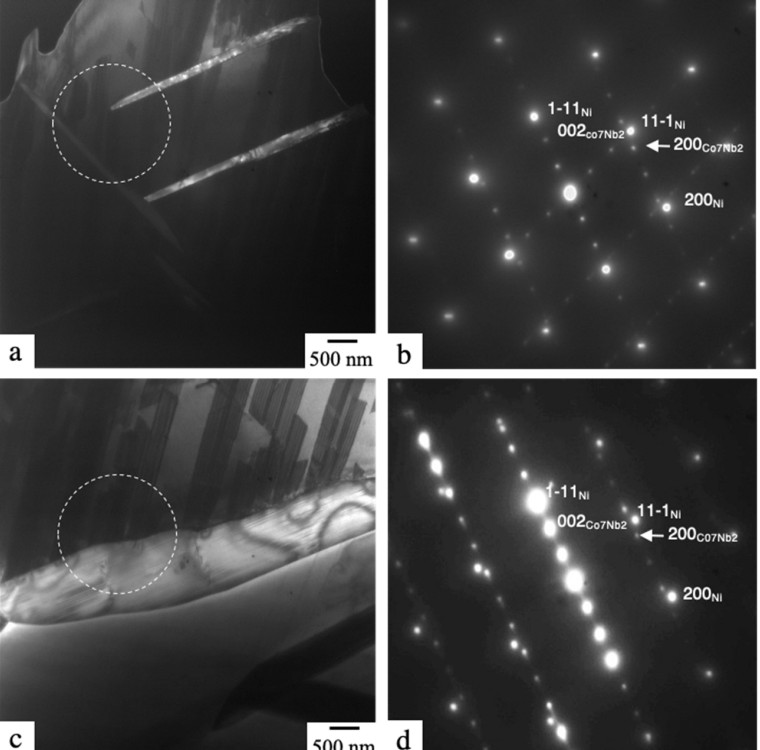

**Figure 2.** TEM images of Ni-5Nb-80Co after solution treatment followed by aging at 1273 K for 100 h: (**a**) dark-field image taken with $g = 001_{Co7Nb2}$; (**b**) selected-area diffraction pattern (SADP) corresponding to the area indicated by a dashed line in (**a**); (**c**) bight-field image taken with $B = 110_{Ni}$; and (**d**) SADP corresponding to the area indicated by a dashed line in (**c**).

Figure 3 shows the change in $\bar{\rho}$ of the $Co_7Nb_2$ phase in Ni-5Nb-80Co aged at 1273 K, together with data for the $Ni_3Nb$ phase ($Ni_3Nb$) in Ni-12Nb-3Fe aged at 1423 K [20]. The $\bar{\rho}$ of the $Co_7Nb_2$ phase increased dramatically after nucleation and became greater than 90% after 1 h. Notably, the difference in $\rho$ decreased compared with that for the $Ni_3Nb$ phase and was less than 10%, independently of the aging time; for example, the $\bar{\rho}$ ranged from 91.5% to 100% at 10 h. The difference in precipitation behavior between the TCP phase and the GCP phase was caused by the difference in growth behavior between these phases at the GB; that is, the TCP phase grew along the GB, whereas the GCP phase grew toward the GI on the habit plane depending on the geometry of the habit plane, $\{111\}_{Ni}$, with respect to the GB plane.

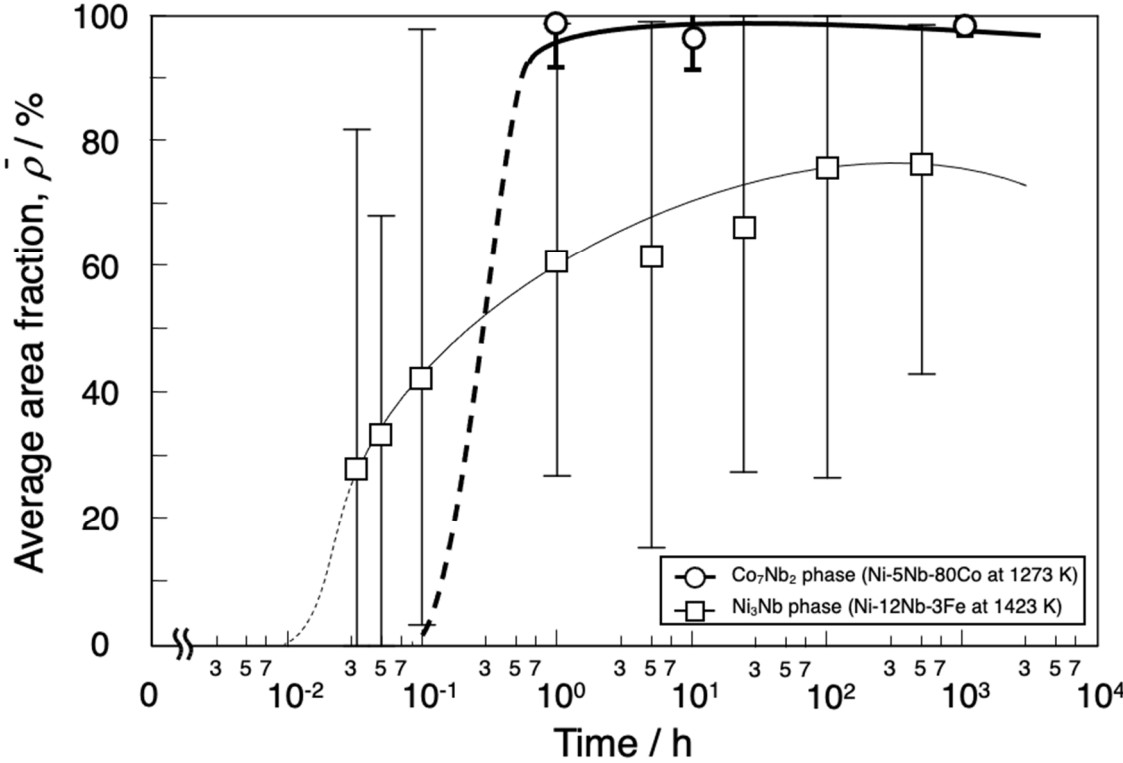

**Figure 3.** Change in the average area fraction of the $Co_7Nb_2$ phase on the GB in Ni-5Nb-80Co aged at 1273 K, together with data for the $Ni_3Nb$ phase in Ni-12Nb-3Fe aged at 1423 K [23].

*3.2. Precipitation of the $Co_7Nb_2$ Phase and $(Ni, Co)_3Nb$ Phase at the GB in the Ni–Nb–Co System (Type I)*

The GCP, TCP and A2 phases all preferentially nucleated at the GB; however, the GCP phase grew toward the GI and the TCP and A2 phases grew along the GB in each two-phase region. The precipitation behavior of the GCP phase and TCP phase at the GB in the Ni/TCP/GCP three-phase region in Type I is shown for Ni-6Nb-57Co.

Figure 4 shows the change in the microstructure of Ni-6Nb-57Co with aging at 1273 K after solution treatment at 1473 K for 4 h. After 12 h, precipitates with granular and plate-like morphologies were observed at the GB and discontinuous precipitation was also observed (Figure 4a). On the basis of the morphology, the granular precipitates that grow along the GB should be the $Co_7Nb_2$ phase, whereas the plate-like precipitates and those formed by discontinuous precipitation should be the $(Ni, Co)_3Nb$ phase. After 500 h, a similar precipitation tendency at the GB was observed (Figure 4b). Because the growth of the GCP phase toward the GI could not be suppressed, the area fraction at the GB decreased compared with the case where only the TCP phase precipitated. The precipitation behavior of these phases in the three-phase region in Type I was the same as that in each two-phase region.

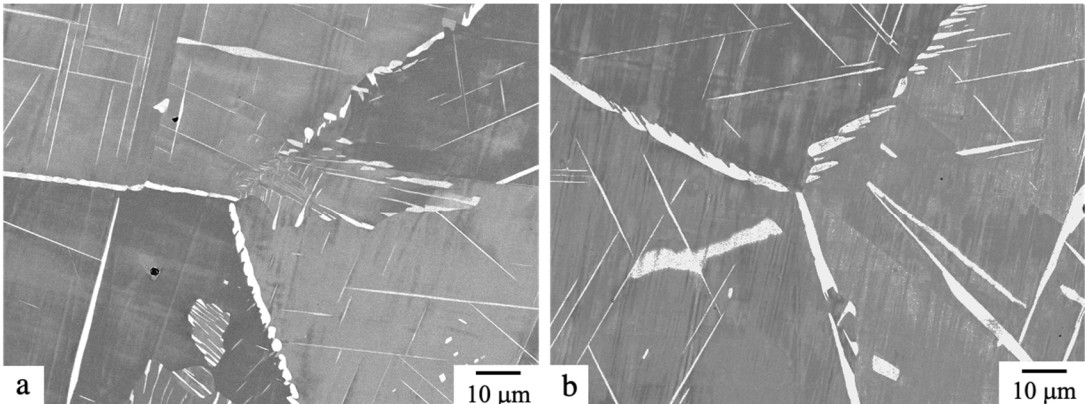

**Figure 4.** Backscattered electron images of Ni-6Nb-57Co after solution treatment followed by aging at 1273 K for (**a**) 12 h and (**b**) 500 h.

### 3.3. Precipitation of Cr and Ni₃Nb Phases at the GB in the Ni–Nb–Cr System (Type II)

The phase diagram of the Ni–Nb–Cr ternary system was investigated to clarify the change in the Ni/Ni₃Nb/Cr three-phase region with temperature. Figure 5 shows the isothermal section of the Ni–Nb–Cr ternary system at 1373 and 1273 K. The solid and dashed lines were drawn on the basis of the composition analysis in the present study and the phase diagram calculated using Pandat 8.1 and PanNi ver. 7, respectively. The results of the phase identification and composition analysis are summarized in Table 1. Ni-15Nb-50Cr is in the Cr/Ni₃Nb/Cr₂Nb (C14 structure) three-phase region at both 1373 and 1273 K. On the other hand, Ni-15Nb-30Cr is in the Ni/Cr/Ni₃Nb three-phase region at both 1373 and 1273 K. The composition of the Ni phase in the Ni/Cr/Ni₃Nb three-phase region changes along equi-Cr composition to the Ni-rich region. On the basis of the analysis results, Ni-3.3Nb-39.4Cr should be in the Ni single-phase region at 1323 K and in the Ni/Cr/Ni₃Nb three-phase region at 1273 K. The precipitation behavior of the GCP phase and A2 phase at the GB in the Ni/A2/GCP three-phase region in Type II is shown using Ni-3.3Nb-39.4Cr.

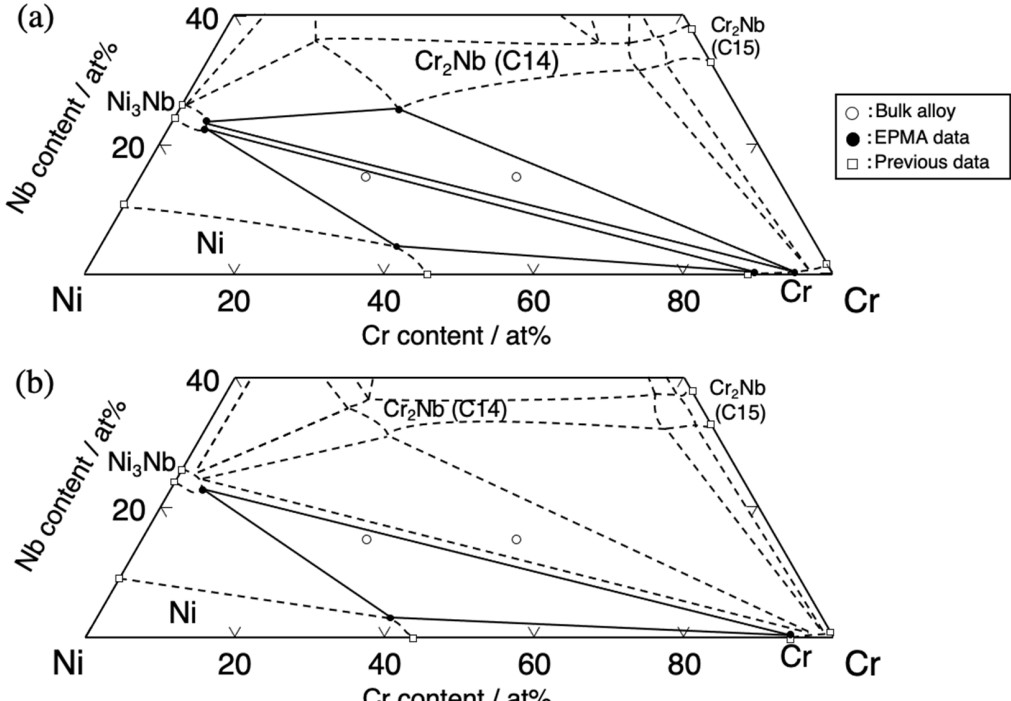

**Figure 5.** Isothermal section of the Ni–Nb–Cr ternary system: (**a**) 1373 K and (**b**) 1273 K.

**Table 1.** Analyzed compositions of the Ni, Cr and Ni$_3$Nb phases in the Ni–Nb–Cr ternary system at 1373 and 1273 K.

| Bulk Alloy Composition | | | Phase Present | Composition (at%) | | | | | |
|---|---|---|---|---|---|---|---|---|---|
| | | | | 1373 K | | | 1273 K | | |
| **Ni** | **Nb** | **Cr** | | **Ni** | **Nb** | **Cr** | **Ni** | **Nb** | **Cr** |
| 55 | 15 | 30 | Ni | 56.1 | 4.3 | 39.5 | 57.7 | 2.6 | 39.7 |
| | | | Ni$_3$Nb | 72.8 | 22.1 | 5.1 | 73.3 | 22.2 | 4.5 |
| | | | Cr | 10.7 | 0.3 | 89.0 | 5.5 | 0.2 | 94.3 |
| 35 | 15 | 50 | Ni$_3$Nb | 72.1 | 23.0 | 4.9 | | | |
| | | | Cr$_2$Nb (C14) | 45.1 | 25.7 | 29.2 | | | |
| | | | Cr | 5.1 | 0.2 | 94.6 | | | |

Figure 6 shows the change in the microstructure of Ni-3.3Nb-39.4Cr aged at 1273 K after the specimen was solution-treated at 1323 K for 12 h. After 1 h of the aging treatment, the Ni$_3$Nb phase with bright contrast and the Cr phase with dark contrast were observed at the GB (Figure 6a). The Ni$_3$Nb phase precipitated independently; however, the Cr phase always precipitated at the interface of the Ni$_3$Nb phase, which means that the Ni$_3$Nb phase precipitates before the Cr phase. After 10 h of solution treatment, both phases grew along the GB. Notably, growth of the Ni$_3$Nb phase toward the GI was not observed (Figure 6b). After 100 h, almost all of the GB was covered by these phases (Figure 6b). After 1000 h, although coarsening occurred and the area fraction decreased, growth of the Ni$_3$Nb phase toward the GI was still suppressed (Figure 6b). The GB precipitation behavior of the GCP phase in the three-phase region in Type II differs completely from that of the Ni/GCP two-phase region [23].

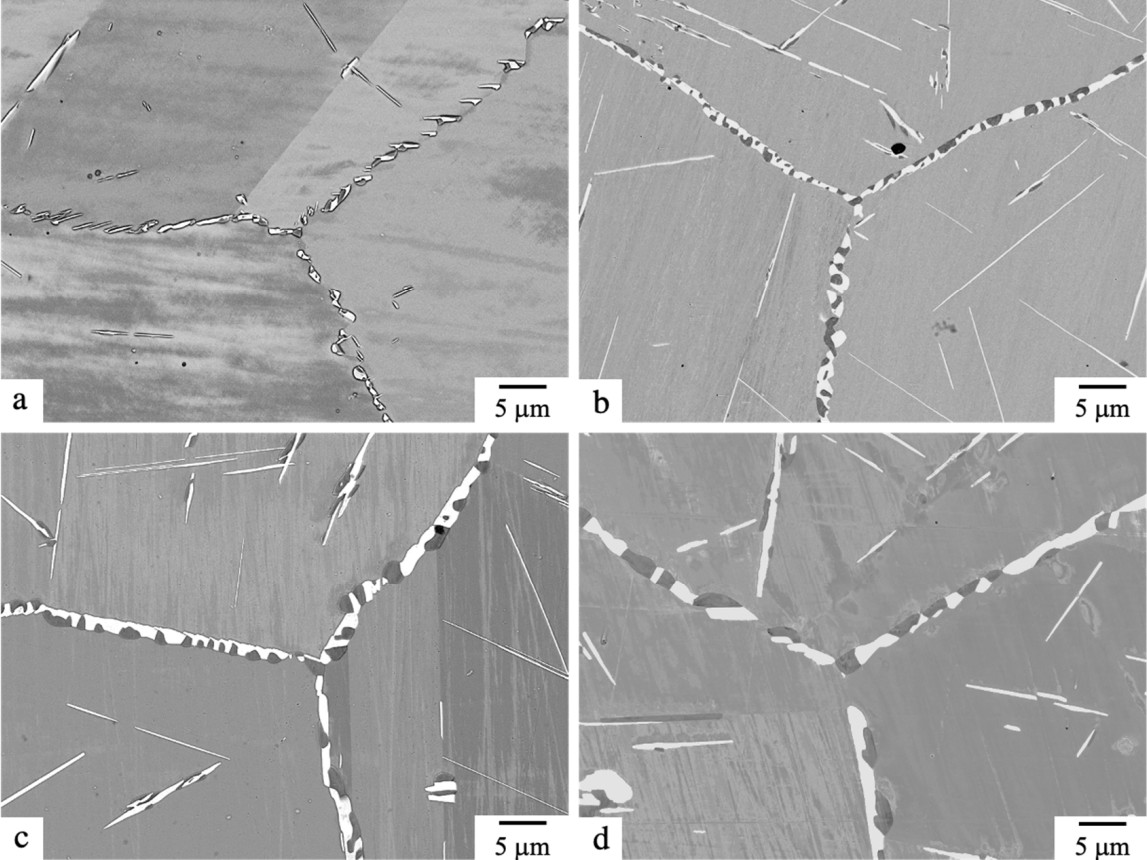

**Figure 6.** Backscattered electron images of Ni-3.3Nb-39.4Cr solution treated and then aged at 1273 K for (**a**) 1 h, (**b**) 12 h, (**c**) 100 h and (**d**) 1000 h.

Figure 7 shows the change in $\bar{\rho}$ of the Ni$_3$Nb phase and the Cr phase at the GB in Ni-3.3Nb-39.4Cr with aging at 1273 K, together with data for the Ni$_3$Nb phase in Ni-12Nb-3Fe aged at 1423 K [20]. The $\bar{\rho}$ gradually increased after nucleation, became greater than 90% at 100 h and then decreased gradually thereafter. The difference in $\rho$ decreased with aging and became approximately 10% after 100 h. The difference in $\rho$ was much smaller than that of the Ni/Ni$_3$Nb two-phase region. These results are attributed to suppression of the growth of the GCP phase toward the GI.

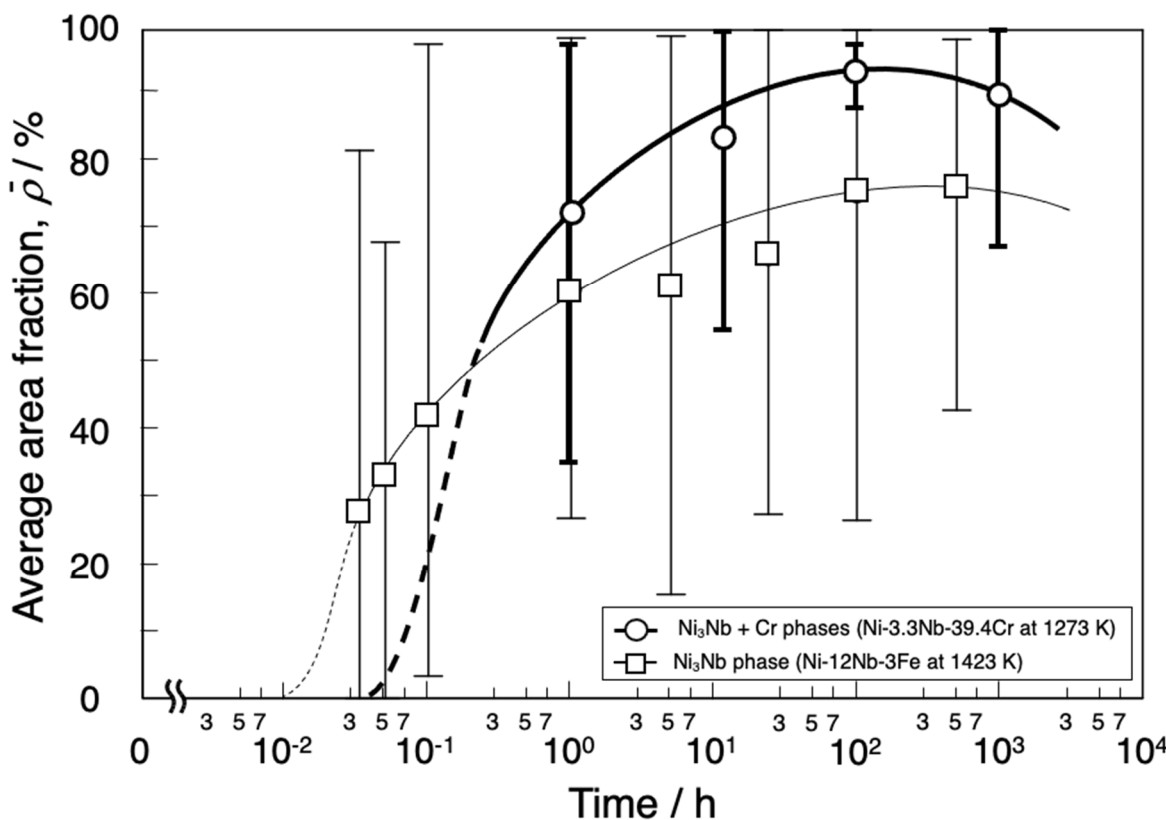

**Figure 7.** Change in the average area fraction of the Ni$_3$Nb phase and the Cr phase on the GB in Ni-3.3Nb-39.4Cr aged at 1273 K, together with data for the Ni$_3$Nb phase in Ni-12Nb-3Fe aged at 1423 K [23].

## 4. Discussion

Growth of the GCP phase toward the GI cannot be suppressed in Type I but can be suppressed in Type II. The effect of precipitation of the TCP or A2 phase on the growth of the GCP phase would be different in Types I and II. The supersaturation for precipitation of the TCP or A2 phase after nucleation of the GCP phase, which occurs prior to nucleation of the TCP or A2 phase, is discussed below.

Figure 8 is a schematic of the shift in composition of the Ni phase as a result of nucleation of the GCP phase from a bulk composition (black symbol) to a non-equilibrium composition (gray symbol), which is a composition outside the Ni/TCP or A2/GCP three-phase region. Lines 1 and 2 show the non-equilibrium tie line and the extended line of the phase boundary between the Ni and GCP phases, respectively. When the GCP phase nucleates, the Ni phase at the interface with the GCP phase becomes a non-equilibrium composition under the assumption that precipitates always have an equilibrium composition. The non-equilibrium tie line of the Ni/GCP two-phase region (line 1) can be drawn by connecting the terminal composition of the GCP phase and the bulk composition. The non-equilibrium composition of the Ni phase would be the intersection of the non-equilibrium tie line (line 1) and the extended line of the phase boundary between the Ni phase and GCP phase (line 2).

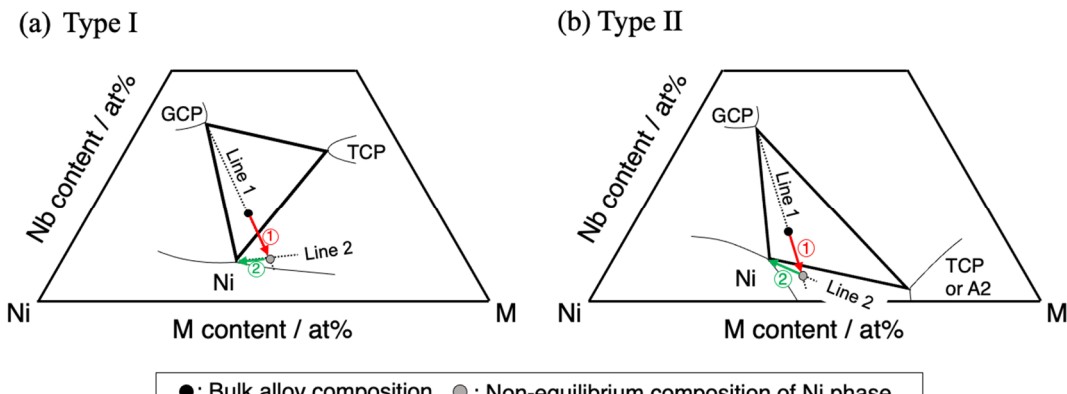

**Figure 8.** Schematic showing the composition shift in the Ni phase from the bulk composition (black symbol) to a non-equilibrium composition (gray symbol) as a result of precipitation of the GCP phase: (**a**) Type I and (**b**) Type II. Line 1 shows the non-equilibrium tie line that connects the terminal composition of the GCP phase and the bulk composition. Line 2 is the extended phase boundary between the Ni and GCP phases. The intersection of these lines is the non-equilibrium composition of Ni phase at the interface with the GCP phase, which precipitates on the GB prior to the TCP or A2 phase.

In Type I (Figure 8a), the non-equilibrium composition of the Ni phase has smaller supersaturation for precipitation of the TCP phase compared with the bulk composition. Therefore, after nucleation of the GCP phase, the supersaturation is used not for nucleation of the TCP phase but rather for growth of the GCP phase toward the GI. Growth of the GCP phase toward the GI occurs prior to nucleation of the TCP phase.

In Type II (Figure 8b), the non-equilibrium composition of the Ni phase could have greater supersaturation for precipitation of the TCP or A2 phase compared with the bulk composition. After nucleation of the GCP phase, nucleation of the TCP or A2 phase occurs prior to growth of the GCP phase toward the GI, which enables the growth of the GCP phase in Type II to be suppressed.

Therefore, precipitation control for covering the GB using both the TCP or A2 and GCP phases might be possible in a system where the precipitation of the GCP phase nucleating on the GB prior to the TCP or A2 phase increases supersaturation for precipitation of the TCP or A2 phase.

## 5. Conclusions

The precipitation behavior of the TCP or A2 phase and the GCP phase at the GB was investigated to clarify the GB design principle using these phases. The conclusions are summarized as follows:

1. In the Ni/Co$_7$Nb$_2$ two-phase region in the Ni–Nb–Co ternary system, the Co$_7$Nb$_2$ phase grows along the GB. The average area fraction of the GB ($\bar{\rho}$) by the Co$_7$Nb$_2$ phase in Ni-5Nb-80Co at 1273 K becomes greater than 90% in 1 h, and the difference in average area fraction becomes less than 10% even at 1000 h. In the Ni/Co$_7$Nb$_2$/(Ni, Co)$_3$Nb three-phase region, however, $\bar{\rho}$ decreases because the (Ni, Co)$_3$Nb phase grows toward the GI.

2. In the Ni/Cr/Ni$_3$Nb three-phase region in the Ni–Nb–Cr ternary system, the $\bar{\rho}$ of Ni-3.3Nb-39.4Cr at 1273 K by the Cr phase and Ni$_3$Nb phase becomes greater than 90% because growth of the Ni$_3$Nb phase toward the GI is suppressed and the GCP phase also grows along the GB.

3. The Ni phase at the interface with the GCP phase becomes a non-equilibrium composition by precipitation of the GCP phase, which precipitates prior to the TCP or A2 phase. The suppression of growth of the GCP phase toward the GI and precipitation control for covering the GB using both the TCP or A2 and GCP phases might be

possible in a system where the precipitation of the GCP phase nucleating on the GB prior to the TCP or A2 phase increases supersaturation for precipitation of the TCP or A2 phase.

**Author Contributions:** Conceptualization, M.T.; Formal analysis, S.I.; Funding acquisition, M.T.; Investigation, S.I.; Resources, M.T.; Supervision, S.K. and M.T.; Writing—original draft, S.I.; Writing—review & editing, R.Y., H.N., S.K. and M.T. All authors have read and agreed to the published version of the manuscript.

**Funding:** This research was financially supported by the Cross-ministerial Strategic Innovation Promotion Program (Development of production technology for high temperature materials designed for jet engines through innovative processes) by the Cabinet Office, Government of Japan.

**Institutional Review Board Statement:** Not applicable.

**Informed Consent Statement:** Informed consent was obtained from all subjects involved in the study.

**Data Availability Statement:** Not applicable.

**Conflicts of Interest:** The authors declare no conflict of interest.

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
