# Peer review of "Grain Boundary Precipitation Control of GCP Phase Using TCP or A2 Phase in Ni-Based Alloys"

_metals, doi:10.3390/met12111817_

Round 1

Reviewer 1 Report

In this manuscript, Grain-boundary precipitation was studied in Ni-based alloys. The work presented here is reasonable and can be considered for publication after a revision.

1. The Abstract is confusing. It would be proper to modify this section.

2. How reliable are the data presented in figures 3 and 7? Considering error bars.

3. Figure 5 needs to be more clearly explained.

4. The references are old. Please use more updated references

Author Response

Thank you very much for valuable and instructive comments. 

We carefully read the manuscript and made some corrections along with your comments.

Reviewer 2 Report

In this article the authors discuss work involving the grain boundary precipitation behavior of topologically close-packed and geometrically close-packed phases. Their findings include the result that the growth of the latter phase toward the grain interior can be suppressed in the A1/A2 three-phase region in the Ni-Nb-Cr ternary system. A decent article that is well put together although I have a few comments.

1. How did the authors determine the size of the particles used in the X-ray diffraction characterization?

2. Please provide references for the equations used.

3. In Figures 3 and 7, please zoom out on the graph to fully show the error bars.

4. In the Conclusions section, please better explain how your work resulted in an advance.

Author Response

(The authors gave the same response as above.)

Reviewer 3 Report

The well written manuscript deals with the very interesting topic of how to control the supersaturation-driven precipitation and growth of different types of phases on grain boundaries of Ni(Co)-based model alloys. The topic fits perfectly well to ‘Metals’ and the article is definitely worth to be published. However, there are some mandatory changes/improvements that have to be done before:

Abstract: The meaning of „M“ and "Type I" and "Type II" must be explained as the Abstract is an independent text and otherwise not understandable.

The authors use generalized formulations such as “Type I” and “Type II” systems, and “M1” and “M2” elements instead of writing the actual systems and metals. This gives the impression that the results have general character for many different systems and metals. However, there is no mention of either other systems or other metals than those studied. Of course the reader wants to know which other elements might be “M1” and “M2” and which other systems belong to the categories “Type I” and “Type II”, otherwise the introduction of these categories makes no sense.

The use of different mixed nomenclatures for designating the phases is somewhat confusing. Some phases are denoted by their space group, others by the Strukturbericht designations. Why not use the original phase names? The current nomenclature might be misunderstood in a sense that these observations are true for all kind of phases belonging to the respective space group or Strukturbericht group. Is it the intention of the authors to state that their observations would be true for any kind of phase belonging to these groups? I would doubt that. The situation becomes even more confusing when as a third category the terms GCP and TCP phases are used. For example when talking about “the A1 and the GCP phase” in one sentence, because actually A1 of course is also a GCP phase.

The presented microstructure design concept is based on the growth of phases along the GB (instead of only precipitating in the GI). So, first and most important question is how that can be controlled and how does one know which phases precipitate at the GB and not in the GI. The authors state that all TCP and A2 phase nucleate and grow along the GBs, but why should that be? This is a central point of the concept. If the driving force for phase precipitation is the supersaturation of the A1 matrix, then one would expect to find the precipitates in the GI. To form the precipitates at the GBs, the atoms from the GI have to diffuse huge distances, which one never would expect to be the preferred mechanism. Perhaps the text in lines 96-99 is intended to give an answer, but I do not understand the text. What do the authors mean by 'regularly take in atoms'? The structure of TCP phases is 'close-packed' and dense, there is no place to 'take in' more atoms (of course excluding interstitial atoms). Most likely I misunderstood the meaning of the authors' text. I suggest to rewrite this part in order to make this important point more clear.

The following paragraph deals with the different kinetics of phase precipitation. The conclusion (lines 104-106) is not clear. Does it mean the authors now expect ‘TCP and A2’ phase precipitation at the GBs only if simultaneously also the GCP phase precipitates at the GB?

Fig. 3: There are 3 points for the mC18 phase all of them >95%. How could the authors draw the dashed line (0% at 0.1h and 95% at 1 h)? Why, for example, isn't this drop either shifted to another interval to the left, or alternatively, is much smoother, for example starting already at 0.01h or even before? There is no reason to give this curve that steep slope.

Lines 266-269: “When the GCP phase nucleates, the A1 phase at the interface with the GCP phase becomes a nonequilibrium state under the assumption that precipitates always have an equilibrium composition.”: This is a somewhat strange way to describe the situation. Actually, the A1 phase does not BECOME a nonequilibrium phase due to the precipitation, but instead it is just the other way round: Only because the A1 phase IS a nonequilibrium (i.e. supersaturated) phase, precipitation starts. This also means that due to the beginning precipitation, the A1 phase starts moving in the direction of equilibrium (what is just the opposite of becoming now a nonequilibrium phase due to GCP phase nucleation). It is very difficult to follow the authors’ argumentation with the help of Fig. 8. The situation might become easier to understand if the figure would show how the positions of the non-equilibrium compositions change with time.

Only in Fig. 8, the special designations “M1” and “M2” are used, but do not occur anywhere else in the text.

If Fig. 8 (a) is intended to show the “Type I” system represented by the Ni-Nb-Co system, then this figure is very misleading. The ‘GCP’ phase marked on the left boundary would be the D019 phase (Ni,Co)3Nb, which is not a binary phase but a purely ternary phase according to the authors preceding publication about this system (Ref.25). This means the left axis is not the Ni-M1 axis but is located somewhere in the middle of the ternary system.

Lines 110-111: Nb ... forming element for A2? Not clear. Are you talking about bcc-Nb? A2 in the later text of the manuscript refers to bcc-Cr, not to bcc-Nb. In addition: the expression ‘forming element’ is not a good choice. For example, adding 25 at.% Nb to Ni results in the formation of Ni3Nb, but this phase is formed by Ni and Nb together, i.e., both are the ‘forming elements’, Ni3Nb also forms by adding 75 at.% Ni to Nb.

Lines 127-128: One cannot say anything about positions of alloy/phase compositions at 1273 K as the respective phase equilibria are not known, Ref. 25 is only for 1373 and 1473 K.

Line 136: What is the meaning of 'diffusion distance'? Should it be 'diffusion coefficients'?

Line 148: What is an 'equilibrated temperature', and which temperature was chosen?

Line 156: “area fractions were measured”: How have these area fractions been measured, i.e., which method was used?

Line 175: The text should read Fig. 1 (d) instead of 1 (b).

Fig. 1 and all others with BSE micrographs: At least in my pdf version, the text at the microbars in all figures contains the infinity sign êš™, must be replaced by µ.

Author Response

(The authors gave the same response as above.)

Round 2

Reviewer 3 Report

Thank you for addressing all points, explaining and answering all questions, and making corresponding changes in the manuscript. I am very sorry, but there remain two minor, but still imortant points that need to be corrected:

- The newly introduced designations ‘α-Cr’ and ‘γ-Ni’ make no sense. I guess the authors want to express that the crystal structure of Cr is bcc and that of Ni is fcc. However, the Greek symbols ‘α’ and ‘γ’ are no synonymous expressions for ‘bcc’ and ‘fcc’. For example, while α-Fe is bcc, α-Co is fcc and α-Ti is hcp. Similar examples can be easily found for ‘γ’ (and structures might be even more complex, e.g., γ-Se is hexagonal (not hcp), hP3, Strukturbericht designation A8). Another reason not to write ‘α-Cr’ and ‘γ-Ni’ is that both Cr and Ni only have this one crystal structure in the solid state (i.e., there is no β-Cr or γ-Cr or anything else, so no need to add a prefix). Therefore it should simply be ‘Cr’ and ‘Ni’.

- There is still the statement that “The planes of the TCP phase … regularly take in atoms from the close-packed plane”. I thought again longer about what could be meant by it, but still I have no idea. This sentence must be changed as its meaning is totally unclear. What is “regularly take in atoms”?? And which “close-packed plane” are the authors talking about?

Author Response

Thank you very much for valuable and instructive comments. 
